# Adversarially Robust Graph Classification: A Pooling-Based Defense Framework

## Abstract

Graph Neural Networks (GNNs) have shown great success across various domains but remain vulnerable to adversarial attacks. While most defense methodology focuses on node classification and enhancing robustness during training, this work shifts the focus to graph classification and inference-time defenses. We theoretically show that the final pooling operation, that is required for graph-level tasks, can have an impact on the graph classifier's underlying robustness. Based on this analysis, we propose a pre-pooling operation, called R-Pool (Robust-Pooling), which is based a novel filtering mechanism using Gaussian Mixture Models (GMMs) to detect and exclude nodes heavily impacted by attacks, thereby enhancing robustness at inference time. Our framework can be used with any pooling operation and any underlying model, and does not require re-training the model nor adapting its architecture. Our experiments demonstrate that this approach effectively mitigates adversarial effects while maintaining a balance between clean and attacked accuracy. Through extensive evaluations on state-of-the-art adversarial attacks, we show that the proposed framework significantly improves the robustness of the underlying GNNs in graph classification tasks compared to other available post-hoc defense methods.

## 1 Introduction

Graph Neural Networks (GNNs) (Kipf & Welling, 2017; Xu et al., 2019b; Veličković et al., 2018) have emerged as a robust framework for learning representations of nodes and graphs, demonstrating notable success across a wide range of real-world applications. These models, which generalize neural network architectures to handle graph-structured data, have been successfully applied in critical domains such as protein function prediction (Kearnes et al., 2016), antibiotic resistance prediction (Qabel et al., 2022), session-based recommendation systems (Wu et al., 2019b) and lately tabular data (Alkhatib et al., 2023). Despite their success, recent studies (Günnemann, 2022) have highlighted the vulnerability of GNNs to adversarial perturbations, small and deliberately introduced changes in the adjacency matrix or node features that can lead to incorrect predictions. These attacks pose significant challenges for the reliable deployment of GNNs, particularly in critical sectors such as healthcare. In response, a growing body of research has focused on characterizing these vulnerabilities through various adversarial attack strategies (Dai et al., 2018; Zügner et al., 2018), while simultaneously advancing defense mechanisms (Wu et al., 2019a; Zhang & Zitnik, 2020) to mitigate these risks and improve the robustness of GNN models.

Existing research predominantly focuses on node-classification tasks, with comparatively limited attention given to robustness analysis in the context of graph-classification tasks (Jin et al., 2021). Although both node-level and graph-level tasks rely on the message-passing propagation mechanism (Gilmer et al., 2017), graph-level tasks introduce additional complexity, requiring the generation of a global graph representation for input graphs of varying sizes and topologies. In such tasks, the pooling mechanism (Cai et al., 2021; Duvenaud et al., 2015) plays a critical role, as it consolidates the node representations obtained from the propagation step into a smaller graph or a single vector. Pooling operations generally fall into two main categories (Liu et al., 2022). The first, hierarchical pooling (Cai et al., 2021), incrementally reduces the graph size, ultimately producing the graph representation for downstream tasks. The second, flat pooling (Duvenaud et al., 2015), directly constructs a graph-level representation in a single step by aggregating node representations, as seen in techniques like sum-pooling. This work specifically focuses on the latter category, ex-

amining commonly used flat pooling operations such as sum, average, and max pooling, which are favored in the graph literature for their simplicity and effectiveness.

While most existing approaches to adversarial defense concentrate on modifying the message-passing scheme—through mechanisms such as attention (Zhang & Zitnik, 2020), adjusting weights (ABBAHADDOU et al., 2024) or pre-processing the adjacency matrix (Wu et al., 2019a), our work takes a different approach by theoretically investigating the impact of the pooling operation on the overall robustness of the model. Specifically, we aim to understand how the choice of pooling method influences adversarial robustness. Additionally, the majority of methods focus on enhancing robustness during the training phase, with relatively few addressing robustness at inference time. This latter perspective is becoming increasingly important with the growing interest in foundation models, where the model is pre-trained, and the goal is to adapt it to downstream tasks while ensuring its adversarial resilience. To the best of our knowledge, this work is the first to explore the effect of pooling operations on the adversarial vulnerability of models in graph classification, specifically at inference time.

In this work, we begin by introducing the concept of adversarial attacks in the context of graph-based models, followed by a formalization of a graph classifier's robustness. This formal framework, which derives an upper bound on the model's expected adversarial risk, enables us to quantify the model's vulnerability within a defined neighborhood. We then apply this formalization to conduct a theoretical analysis of the robustness of various pooling operations, extracting their corresponding upper bounds. Building on these theoretical insights, we propose *Robust-Pooling ("R-Pool")*, an incremental component that can be integrated with any existing pooling operation to enhance the model's robustness at inference time. Our theoretical findings suggest that certain nodes in the graph are more prone to accumulating adversarial perturbations in their final representations. To address this, the proposed framework aims to mitigate the impact of these perturbations by filtering out such nodes. Specifically, the proposed method consists of fitting a Gaussian Mixture Models (GMMs) to node representations and using an out-of-distribution score (Morteza & Li, 2022) to detect nodes heavily impacted by adversarial attacks, excluding them prior to the pooling operation. Although this filtering may lead to a loss of information for the downstream classification task, our experiments demonstrate that with an appropriate threshold, a balance between clean and adversarial accuracy can be achieved. Finally, we validate our theoretical contributions through extensive experimental evaluations, using state-of-the-art graph-classification adversarial attacks. The results confirm the validity and practical value of the proposed framework. The contributions of this work can be summarized as follows:

- We theoretically analyze how various pooling operations impact a model's underlying robustness in the case of graph classification.

- We then introduce "R-Pool", a novel filtering mechanism, based on fitting Gaussian Mixture Models (GMMs) and using an out-of-distribution score to detect and exclude nodes heavily impacted by adversarial attacks, thereby enhancing robustness during inference.

- We validate experimentally the efficiency of the proposed framework, demonstrating improved robustness against adversarial attacks in graph-classification tasks while maintaining a balance between clean and adversarial accuracy.

## 2 RELATED WORK

There has been growing interest in adversarial attacks targeting Graph Neural Networks (GNNs) Zügner et al. (2018); Dai et al. (2018). Various attack methods have been proposed depending on the attack setting, which may assume full access to the model's architecture and training data (white-box) or limit the attacker to model queries or surrogate models (black-box). Most attacks frame the problem as an optimization task, with solutions ranging from PGD-based methods (Xu et al., 2019a) to meta-gradient approaches (Zügner & Günnemann, 2019) and reinforcement learning strategies (Dai et al., 2018). While much of the literature focuses on node classification, with some methods like PGD adaptable to graph classification, few approaches are specifically tailored to this task. For instance, Wan et al. (2021) employ Bayesian optimization, and Dai et al. (2018) explore several strategies, including reinforcement learning, gradient-based attacks, and genetic algorithms, where graph modifications evolve through a population-based approach guided by a fitness function.

On the defense side, recent efforts have emerged to protect GNNs from adversarial attacks. Some approaches focus on pre-processing the adjacency matrix to improve robustness. For example, Wu et al. (2019a) use Jaccard similarity to filter out adversarial edges, while Entezari et al. (2020) apply SVD decomposition to denoise the adjacency matrix before feeding it into the model. Other defenses target the structure of the GNN itself. Zhang & Zitnik (2020) introduce edge pruning through an attention mechanism to remove vulnerable connections, while ABBAHADDOU et al. (2024) and Ennadir et al. (2024) propose modifications to the message-passing scheme to reduce the impact of adversarial perturbations. Moreover, adversarial training methods, such as those presented by Gosch et al. (2024), have been developed to increase the model's resilience by explicitly training the GNN with adversarially perturbed graphs. Additionally, a growing interest in exploring robustness certificates (Zügner & Günnemann, 2019; Bojchevski & Günnemann, 2019) have emerged such as randomized smoothing (Bojchevski et al., 2020). However, most of these defense techniques are tailored to node classification tasks and have limited direct application to graph classification.

Our work takes a different direction by focusing specifically on the graph classification task, which has received comparatively less attention in adversarial defense literature. In particular, we investigate the role of pooling operations in determining the robustness of GNNs. Pooling, a key operation in graph classification models, has not been thoroughly studied in terms of its influence on adversarial vulnerability, especially during inference time. This paper aims to fill that gap by offering a theoretical analysis of how pooling affects model robustness. To our knowledge, this is the first work to explore this aspect in the context of graph classification, contributing a novel perspective to the ongoing research in adversarial attacks on GNNs.

## 3 PRELIMINARIES

Before continuing with our contribution, we introduce some fundamental concepts and notations.

**Notation and Setup.** Let $G = (V, E)$ be a graph where $V$ is its set of vertices and $E$ its set of edges. We denote by $n = |V|$ and $m = |E|$ the number of vertices and number of edges, respectively and $\mathcal{N}(v) = \{u \colon (v, u) \in E\}$ the set of neighbors of a node $v \in V$. The degree of a node is equal to its number of neighbors, i. e., $|\mathcal{N}(v)|$ for a node $v \in V$. A graph is commonly represented by its adjacency matrix $\mathbf{A} \in \mathbb{R}^{n \times n}$, where the $(i, j)$-th element of the adjacency matrix is equal to the weight of the edge between the $i$-th and $j$-th node of the graph and a weight of 0 in case the edge does not exist. In some settings, the nodes of a graph might be annotated with feature vectors. We use $\mathbf{X} \in \mathbb{R}^{n \times D}$ to denote the node features where $D$ is the feature dimensionality. The feature of the $i$-th node of the graph corresponds to the $i$-th row of $\mathbf{X}$.

**Message Passing GNNs.** A GNN model consists of a series of neighborhood aggregation layers which use the graph structure and the nodes' feature vectors from the previous layer to generate new representations for the nodes. Specifically, GNNs update nodes' feature vectors by aggregating local neighborhood information. Suppose we have a GNN model that contains $T$ neighborhood aggregation layers. Let also $\mathbf{h}_v^{(0)}$ denote the initial feature vector of node $v$, i. e., the row of matrix $\mathbf{X}$ that corresponds to node $v$. At each iteration ($t > 0$), the hidden state $\mathbf{h}_v^{(t)}$ of a node $v$ is updated as follows:

$$\mathbf{a}_v^{(t)} = \text{AGGREGATE}^{(t)}\left(\left\{\mathbf{h}_u^{(t-1)} \colon u \in \mathcal{N}(v)\right\}\right)$$

$$\mathbf{h}_v^{(t)} = \text{COMBINE}^{(t)}\left(\mathbf{h}_v^{(t-1)}, \mathbf{a}_v^{(t)}\right),$$

where AGGREGATE is a permutation invariant function that maps the feature vectors of the neighbors of a node $v$ to an aggregated vector. This aggregated vector is passed along with the previous representation of $v$ (i. e., $\mathbf{h}_v^{(t-1)}$) to the COMBINE function which combines those two vectors and produces the new representation of $v$.

**Pooling Operation.** After $T$ iterations of neighborhood aggregation, to produce a graph-level representation, GNNs apply a permutation invariant readout function to the feature vectors of all nodes of the graph as follows:

$$\mathbf{h}_G = \text{READOUT}\left(\left\{\mathbf{h}_v^{(T)} \colon v \in V\right\}\right),$$

Our study focuses on the Flat Pooling Family which directly generated a graph-level representation in one step (e. g., sum operator, mean operator and max operator).

# 4 ON THE ROBUSTNESS OF POOLING OPERATIONS

This work focuses on the robustness of the graph classification task within the broader context of graph representation learning, with a particular emphasis on understanding the impact of pooling operations on model robustness. We begin by discussing the concept of robustness in graph classifiers, followed by a formal mathematical definition of model robustness. Finally, we provide theoretical insights into the robustness of widely-used flat pooling operations, offering a deeper understanding of their behavior under adversarial attacks.

## 4.1 ADVERSARIAL GRAPH ROBUSTNESS

Given an input graph (with its corresponding node attributes), graph-level classification aims to learn a function that predicts a property of interest related to the graph. Let's therefore consider our set of input graphs $\{(G_1, X_1, y_1), \ldots, (G_N, X_N, y_N)\} \in (\mathcal{G}, \mathcal{X}, \mathcal{Y})^N$ considered to be sampled from an underlying distribution $\mathcal{D}$ defined on $(\mathcal{G}, \mathcal{X}, \mathcal{Y})$. Graph classification aims to find the graph-classifier $f : (\mathcal{G}, \mathcal{X}) \to \mathcal{Y}$ minimizing the classification risk w.r.t $\mathcal{D}$, which is defined as:

$$R[f] := \mathbb{E}_{(G,X,y)\sim\mathcal{D}}[\mathbf{1}\{f(G, X) \neq y\}].$$

In this work, we focus on the black-box evasion attack setting, where we consider that the user cannot access/modify the trained and static victim model $f : (\mathcal{G}, \mathcal{X}) \to \mathcal{Y}$ or the training dataset. For our theoretical insights, we follow the same definition as the one provided in the work, by adapting it to the case of graph-classification. Specifically, let's consider an input graph $(G, X) \in (\mathcal{G} \times \mathcal{X})$ with its corresponding label $y \in \mathcal{Y}$, an adversarial attack aims to degrade the performance of the considered victim model by finding a graph $\tilde{G}$ and its corresponding features $\tilde{X}$ within the the input graph's neighborhood for which the predicted classification is different from the original classification of the consider graph input $(G, X)$. In this perspective, assessing the adversarial risk of a classifier consists of analyzing the input graph's neighborhood by quantifying the "chances" (in terms of expectancy) of finding an adversarial graph that is similar to input graph and for which the model's output is different than the original output. For a given input graph's neighborhood defined by a threshold $\epsilon$, this expected risk robustness quantification can be formulated as follows:

$$\mathcal{R}_\epsilon[f] = \mathbb{E}_{\substack{(A,X)\sim\mathcal{D} \\ (\tilde{A},\tilde{X})\in\mathcal{N}_\epsilon(A,X)}} [d_{\mathcal{Y}}(f(\tilde{A}, \tilde{X}), f(A, X))], \tag{1}$$

where we consider $d_{\mathcal{Y}} = \|.\|_2$ as a distance within our output space $\mathcal{Y}$. Additionally, $\mathcal{N}_\epsilon(A, X) = \{(\tilde{A}, \tilde{X}) : d_{\mathcal{A},\mathcal{X}}([A, X], [\tilde{A}, \tilde{X}]) < \epsilon\}$ denotes the input graph's considered neighborhood defined by our attack/perturbation budget $\epsilon$. We simply consider the following graph distance that reflects both the structure and the node features:

$$d_{\mathcal{A},\mathcal{X}}([A, X], [\tilde{A}, \tilde{X}]) = \min_{P\in\Pi}\{\|A - P\tilde{A}P^T\|_2 + \|X - P\tilde{X}\|_2\},$$

with $\Pi$ being the set of permutation matrices. In the case of un-attributed graphs, the previous distance resolves to using only the first part related to the adjacency matrices.

Typically, from the previous formulation, from a defense perspective, we aim to have the smallest possible value, meaning that within the considered perturbation budget, the distance in term of output of the attacked graph and the clean graph is not very big. Hence, we would expect the two of them to have the same classification, which reflects the failure of the attack. Deriving the precise expected distance, as defined in Equation (1), is challenging. However, an effective and more manageable approach is to establish an upper-bound on this risk. By deriving such upper-bound, users can gain a comprehensive understanding of the GNN's susceptibility to adversarial attacks and make informed assessments of its robustness based on the specific task at hand. For example, in certain scenarios like social networks, where a limited number of successful attacks may not have severe consequences, a larger upper-bound on the adversarial risk might be tolerable. While in other more sensitive areas, such as financial applications, we need to aim for a much tighter upper-bound to

control the confidence level of the adversarial risk. From this perspective, Definition 4.1 introduces the notion of a GNN's robustness.

**Definition 4.1.** (Adversarial Robustness). The graph-based function $f : (\mathcal{A}, \mathcal{X}) \to \mathcal{Y}$ is said to be $(\epsilon, \gamma) -$ robust if its adversarial risk is upper-bounded i.e., $\mathcal{R}_\epsilon[f] \leq \gamma$ with respect to the chosen graph distances in the input and output measurable sets.

As previously precised by the work, the definition rather approaches the adversarial problem from an "average perspective", where we analyze the whole neighborhood rather than "worst-case", in which the focus is on a single adversary that results in the most harmful performance. Note that, *if f is $(\epsilon, \gamma) -$ "robust", then it is also $(\epsilon, \gamma) -$ "worst-case robust"*. The complete difference between worst-case and average case have been thoroughly studied previously in the literature.

## 4.2 ON THE ROBUSTNESS OF FLAT POOLING OPERATIONS

Having established a formal definition and framework for robustness in the case of graph classification, we now apply this framework to examine the robustness of commonly used pooling operations. In this context, we focus on two well-established message-passing models, which are instances from the general perspective provided in Section 3 : Graph Convolutional Networks (GCNs) and Graph Isomorphism Networks (GINs).

To improve the quality of graph-level representations, various pooling mechanisms have been proposed in the literature, which can be broadly categorized into Flat Pooling and Hierarchical Pooling. In this study, we focus on three widely used flat pooling methods, typically the Sum, Average, and Max pooling (Duvenaud et al., 2015). These methods directly generate a graph-level representation in a single step by aggregating node embeddings. While our theoretical analysis focuses on these specific pooling operations, we consider that our theoretical analysis can be easily expanded to take into account other available pooling operations within the Flat pooling sub-family.

Our analysis covers both node feature-based adversarial attacks, which modify node attributes to change model predictions, and structural perturbations, where adversaries alter the graph's structure by adding or removing edges to achieve their goal.

**Theorem 4.2.** *Let $f : (\mathcal{G}, \mathcal{X}) \to \mathcal{Y}$ denote a graph-based function composed of $L$ GCN layers, where the weight matrix of the $i$-th layer is denoted by $W^{(i)}$. Further, let $d^{0,1}$ be a graph distance. For adversarial attacks only targeting node features of the input graph, with a budget $\epsilon$, we have:*

- *If f is Max-pooling based classifier, then f is $d^{0,1}$-$(\epsilon, \gamma)$ robust with:*

$$\gamma = \prod_{l=1}^{L} \mid W^{(l)} \mid \max_{u \in \mathcal{V}} \hat{w}_u \epsilon.$$

- *If f is Sum-pooling based classifier, then f is $d^{0,1}$-$(\epsilon, \gamma)$ robust with:*

$$\gamma = \prod_{l=1}^{L} \mid W^{(l)} \mid \sum_{u \in \mathcal{V}} \hat{w}_u \epsilon.$$

- *If f is Average-pooling based classifier, then f is $d^{0,1}$-$(\epsilon, \gamma)$ robust with:*

$$\gamma = \frac{\epsilon}{|V|} \prod_{l=1}^{L} \mid W^{(l)} \mid \sum_{u \in \mathcal{V}} \hat{w}_u,$$

*with $\hat{w}_u$ denoting the sum of normalized walks of length $(L-1)$ starting from node $u$.*

Theorem 4.2 provides the respective upper bounds on the different considered pooling operations. The derived bound depends on two main factors. The first factor relates to the model itself, where we observe the norm of the weights. The second factor is associated with the underlying structure of the input graph. When subject to targeted attacks, we find that the Average and Sum pooling operations consistently demonstrate vulnerability to adversarial manipulation. In contrast, Max pooling exhibits variable robustness. Specifically, it can be more resilient if the targeted attack happens to affect a node that is not the most connected (in terms of walks). However, it can become significantly more vulnerable if the attack targets the most connected node, which is likely in scenarios such as gradient-based attacks, as these focus on nodes with the highest impact. In the case of more global, non-targeted attacks, the effect on Max pooling can be expected to be less pronounced than

on Average and Sum pooling. This is because Max pooling accumulates the effect through only one node, while the others accumulate the effects of all nodes. Overall, our analysis suggests that there isn't a universally more robust pooling operation. Rather, the relative robustness depends on the specific attack setting and the different types of attacks employed. This underscores the importance of considering the anticipated threat model when selecting an appropriate pooling strategy for GNNs in adversarial environments. We note that while this latter study is focusing on GCN, a similar approach can be applied for GINs.

**Theorem 4.3.** *Let $f : (\mathcal{G}, \mathcal{X}) \to \mathcal{Y}$ be composed of $L$ GIN-layers (with its internal parameter $\zeta = 0$) and let $W^{(i)}$ denote the weight matrix of the $i$-th MLP layer. We consider the input node feature space to be bounded i. e., $\|X\|_2 < B$ for some $B \in \mathbb{R}$. For node feature-based attacks, with a budget $\epsilon$, the function $f$ is $(d^{0,1}, \epsilon)$–robust with*

- *If $f$ is Max-pooling based classifier, then $f$ is $d^{0,1}$-$(\epsilon, \gamma)$ robust with:*

$$\gamma = \prod_{l=1}^{L} \|W^{(l)}\| \left[ B \times L \times \max_{u \in V} \deg(u) + \epsilon \right].$$

- *If $f$ is Sum-pooling based classifier, then $f$ is $d^{0,1}$-$(\epsilon, \gamma)$ robust with:*

$$\gamma = \prod_{l=1}^{L} \|W^{(l)}\| \left[ 2B \times L \times |E| + |V|\epsilon \right].$$

- *If $f$ is Average-pooling based classifier, then $f$ is $d^{0,1}$-$(\epsilon, \gamma)$ robust with:*

$$\gamma = \prod_{l=1}^{L} \|W^{(l)}\| \left[ \frac{2B \times L \times |E|}{|V|} + \epsilon \right],$$

*with $| E |$ being the number of edges and $| V |$ the number of nodes.*

Similar to the case of GCN, the computed upper-bounds reveal the impact of the pooling operation on the model's underlying adversarial robustness depending on the input graph's structure. Specifically, for the sum pooling, the perturbation scales with the total number of edges and nodes, indicating that larger and denser graphs are more susceptible to adversarial feature perturbations. In contrast, average pooling normalizes this effect by the number of nodes, potentially reducing the overall sensitivity in larger graphs. In the case of max pooling, it seems that it depends on the maximum node degree, suggesting that graphs with highly connected nodes may be more vulnerable to attacks targeting those nodes. Overall, both in the case of GCN and GIN, the theoretical results show that choosing appropriate pooling strategies can have an effect of the model's robustness.

## 5 ROBUST POOLING THROUGH FILTERING

From Section 4.2, it appears that for different pooling operations, the upper bound is dependent on the underlying graph's structure. This observation aligns with the intuitive understanding of message-passing mechanisms, which propagate information within node neighborhoods. In this context, when an attack is injected into certain nodes, its effect propagates to others. Consequently, nodes that are more "influential" in terms of their interactions with other nodes have a higher probability of being affected by the attack, which in turn impacts nodes within their neighborhood, resulting in a compounded effect.

These insights suggest a direct and straightforward defense method: discarding the more highly connected nodes within the graph to mitigate their cumulative attack effect. Similar approaches have been proposed in the literature, employing pre-processing techniques such as Jaccard Similarity or SVD decomposition. However, these methods have primarily been applied to node classification tasks, where graphs are typically large, and discarding a number of nodes does not necessarily result in significant information loss. In contrast, graph classification often deals with smaller graphs, where highly connected nodes are crucial for the downstream task, and their removal could lead to substantial information loss, equivalent to reduced clean accuracy which is very important since a priori we do not know if the graph is attacked or not. Therefore, finding the right trade-off between clean and attacked accuracy is important.

Given these considerations, we propose an alternative solution based on post-message-passing filtering. Specifically, after propagating the messages (e. g., using convolutions in the case of GCN (Kipf

& Welling, 2017)), the result is a matrix $X \in \mathbb{R}^{n,e}$, where $n$ is the number of nodes and $e$ is the embedding dimension. We suppose that the resulting node representations follow a certain distribution $\mathcal{D}$. Consequently, when subject to adversarial attacks, some of these node representations may exhibit anomalous behavior, which is likely to be out-of-distribution with respect to $\mathcal{D}$. This approach is particularly practical for small graphs, where retaining all nodes ensures that information relevant to the downstream task is preserved through the message-passing scheme.

Given a trained model, which we consider static, we introduce a filtering mechanism prior to the pooling operation. Specifically, for an input graph $G$ with $n$ nodes, we consider its node representation $X$ and aim to partition the set of nodes into two subsets: a set of non-affected nodes $\mathcal{N}$ and a set of affected nodes $\mathcal{M}$. Assuming the distribution $\mathcal{D}$, the first set can be viewed as in-distribution points and the second set as out-of-distribution points. Based on this perspective, we approximate the distribution $\mathcal{D}$ using a finite mixture of $k$ components, in particular with a Gaussian Mixture Model (GMM), which serves as a universal approximator for a wide range of density functions. The process involves estimating the GMM parameters, i.e., the component weight $\eta_j$, the mean $\mu_j$ and covariance matrix $\Sigma_j$, for each $j = 1, \ldots, k$ for each input graph using the node representation $X$, implemented via the Expectation-Maximization (EM) algorithm. After computing the optimal parameter values, we adopt an approach similar to that proposed in (Morteza & Li, 2022), an OOD detection method. For each node $u$ with representation $x_u$, we compute a score, denoted as the Gaussian mixture based energy measurement (GEM), which can be expressed as

$$s(u) = \log \sum_{j=1}^{k} \eta_j \exp(-\frac{1}{2}(x_u - \mu_j)^\top \Sigma_j (x_u - \mu_j)),$$

where $x_u$ represents the node representation of node $u$. Given a threshold value $\lambda$, we employ the GEM score to determine whether a node $u$ has been subjected to an attack, consequently supporting our decision to retain or discard it. Our implementation computes the threshold $\lambda$ based on the quantile values of the calculated scores. This approach has demonstrated superior adaptability compared to a rigid threshold, particularly given the heterogeneity in graph sizes prevalent in our datasets. Formally, we classify a node $u$ as $u \in \mathcal{N}$ based on its score $s(u)$ comapred to $\lambda$. It is noteworthy that (Morteza & Li, 2022) provide theoretical insights into the performance characteristics of the GEM methodology, which is out of this work's scope. We additionally note that while in this work, we have chosen to work with the GEM score, any other OOD score could be used. After processing all the nodes and deciding on the set of nodes to be kept $\mathcal{N}$ and to discard $\mathcal{M}$, we proceed with the classical pooling function to produce the final graph representation. We refer to the proposed filtering scheme as R-pool and as it can be seen from its construction, it is adaptable to any pooling e. g., sum, mean or max.

The primary advantage of our method lies in its model-agnostic nature, operating independently of the underlying Graph Neural Network (GNN) architecture. R-Pool accepts any node representation $X$ as input, regardless of whether it is generated by GCN, (Kipf & Welling, 2017), GIN (Xu et al., 2019b), GAT (Veličković et al., 2018), or any other method. It then performs adaptive node filtering to produce a refined set of node representations. This universality ensures broad applicability across various GNN frameworks without necessitating modifications to the original model architecture or retraining procedures. Our approach is particularly relevant in the context of Foundation Models, where the pre-trained nature of the model makes it prohibitively costly to re-train for adapting the message-passing scheme or implementing adversarial training. By operating at inference time, R-Pool offers a practical solution for enhancing robustness without the computational and resource overhead associated with model re-training.

**Complexity of the Method** The main computational complexity of our method is concentrated in the Expectation-Maximization algorithm used for estimating the GMM's parameters. The EM algorithm is an iterative process comprising two main steps: the Expectation (E) step and the Maximization (M) step. In the E-step, for each data point, we compute the probability of it being generated by each component of the model, given the current parameter estimates. The M-step then updates these parameters to maximize the expected log-likelihood based on the probabilities calculated in the E-step. The algorithm's convergence rate and the quality of the resulting fit are partially dependent on the number of iterations. In our implementation and experimental results, we observed that a relatively small number of iterations (between 100 and 200) yielded satisfactory results (both in terms of clean and attacked accuracy). A comprehensive time and complexity analysis is provided in Appendix D.

## 6 EXPERIMENTAL RESULTS

In this section, we aim to validate the validity of our proposed R-Pool when subject to adversarial attacks and using real world datasets. We start by give details about the experimental settings we will be following and then we will report and analyze the performance of our proposed approach in comparison to other available methods.

### 6.1 EXPERIMENTAL SETUP

**Datasets.** Consistent with our theoretical analysis, this section focuses on the graph classification task. We base our evaluation on the standard graph dataset derived from bioinformatics (PROTEINS, NCI1) and from social networks (IMDB-BINARY) (Morris et al., 2020). We note that the social network graphs are unlabeled, while all other graph datasets come with either node labels or node attributes. We take those labels/attributes into account when available and we otherwise use the node's degree as its node features. To mitigate the impact of randomness during training, each experiment was repeated 10 times, using the public train/validation/test splits provided in the work (Errica et al., 2020).

**Attacks.** To evaluate the effectiveness of our proposed R-Pool defense method, we consider four adversarial attack strategies: **(i)** *Random Attack*, which performs random searches by randomly adding or deleting edges in the input graph; **(ii)** *Genetic Attack* (Dai et al., 2018), which modifies graph structures using evolutionary computing principles by evolving a population of candidate solutions through selection, crossover, and mutation to generate adversarial examples; **(iii)** *Gradient-Based Attack* (PGD) (Dai et al., 2018), which greedily adds or deletes edges based on the magnitude of gradients computed with respect to the input graph. For all attacks, we set a perturbation budget of $\epsilon = 0.3$, allowing the modification of up to $30\%$ of the edges in the input graph.

**Architecture.** For all our experiments, a 2-layers GCN classifier with identical hyperparameters and activation functions was employed. The models were trained using the cross-entropy loss function, and consistent values for the number of epochs and learning rate were maintained across all analysis with the Adam optimizer (Kingma & Ba, 2015). Further implementation details can be found in Appendix E and the code implementation to replicate our experiments is provided in the supplementary material and would be available on GitHub upon publication.

**Baselines.** As discussed in Section 2 , few methods exist for defending against adversarial attacks on graph classification tasks, particularly at inference time (post-hoc methods). To evaluate our proposed R-Pool defense, we compare it against two main baselines: **(i)** *Pre-processing techniques*, where the adjacency matrix is pre-processed to remove nodes identified as malicious; specifically, we employ the Jaccard similarity approach (Wu et al., 2019a). **(ii)** *Randomized Smoothing* (Bojchevski et al., 2020), which involves adding random noise to the input graph and making predictions based on a majority vote over multiple noisy samples. We evaluate the methods based on two key metrics. First, the *clean accuracy*, which measures the method's performance on unperturbed graphs; this is particularly important for post-hoc methods since, in practice, we may not know whether a graph has been attacked. Second, we consider the *attacked accuracy*.

### 6.2 EXPERIMENTAL RESULTS

Table 6.2 reports the clean and attacked graph classification accuracy of our proposed R-Pool and other considered benchmarks. A primary observation is that R-Pool maintains clean accuracy comparable to the baseline GCN, in contrast to some alternatives such as pre-processing approaches which often compromise clean accuracy. This finding validates our hypothesis discussed in Section 5, where we argued for the filtering after the message-passing scheme rather than before. Based on this approach, our proposed R-Pool preserves the flow of relevant information within node representations, thereby maintaining the relevant elements for the downstream graph classification task.

When subject to adversarial attacks, our proposed framework demonstrates robust performance, often surpassing or matching the considered benchmarks. In the case of the labeled PROTEINS dataset, R-Pool outperforms other baselines in two out of three attack scenarios, showcasing its robustness against various adversarial perturbations. For the NCI1 dataset, R-Pool exhibits performance comparable to randomized smoothing. Interestingly, while the pre-processing technique

Table 1: Clean and Attacked classification accuracy (± standard deviation) of the considered baselines on different benchmark graph classification dataset when subject to adversarial attacks. The best accuracy in each setting and each dataset is typeset in **bold**.

| Attack | Dataset | PROTEINS | NCI1 | IMDB-BINARY |
|---|---|---|---|---|
| Clean | GCN | $73.2 \pm 0.4$ | $64.6 \pm 0.3$ | **$58.9 \pm 0.2$** |
| | + Random Smoothing | $74.7 \pm 0.3$ | **$64.9 \pm 0.5$** | $53.3 \pm 0.5$ |
| | + Pre-processing | $71.9 \pm 0.7$ | $60.1 \pm 0.4$ | - |
| | + Ours | **$75.1 \pm 0.4$** | $63.7 \pm 0.9$ | $57.4 \pm 0.4$ |
| PGD | GCN | $57.1 \pm 0.8$ | $37.2 \pm 0.3$ | $53.0 \pm 0.4$ |
| | + Random Smoothing | $62.8 \pm 1.3$ | $40.3 \pm 0.6$ | $51.8 \pm 0.7$ |
| | + Pre-processing | $59.3 \pm 0.7$ | **$50.3 \pm 0.2$** | - |
| | + Ours | **$64.6 \pm 1.1$** | $43.6 \pm 0.5$ | **$55.7 \pm 0.9$** |
| Random | GCN | $68.8 \pm 0.9$ | $22.3 \pm 0.6$ | $54.2 \pm 0.3$ |
| | + Random Smoothing | $70.3 \pm 0.8$ | $25.3 \pm 0.7$ | $54.8 \pm 0.6$ |
| | + Pre-processing | $64.3 \pm 0.9$ | **$37.3 \pm 0.4$** | - |
| | + Ours | **$70.6 \pm 1.2$** | $23.1 \pm 0.8$ | **$56.4 \pm 0.8$** |
| Genetic | GCN | $63.4 \pm 0.8$ | $18.4 \pm 0.5$ | $52.4 \pm 0.6$ |
| | + Random Smoothing | **$67.9 \pm 0.7$** | $22.1 \pm 0.4$ | $52.9 \pm 0.7$ |
| | + Pre-processing | $61.7 \pm 0.5$ | **$35.8 \pm 0.5$** | - |
| | + Ours | $67.8 \pm 0.9$ | $22.3 \pm 0.6$ | **$53.2 \pm 0.9$** |

yields the highest attacked accuracy for NCI1, it does so at the cost of significantly reduced clean accuracy, which is a trade-off that may be undesirable in many real-world applications where maintaining performance on unperturbed data is crucial. This observation underscores the importance of considering both clean and attacked accuracy when evaluating defense mechanisms. Additionally, for the unlabeled IMDB-BINARY dataset, where the Jaccard-based method can't be used due to the absence of node features and randomized smoothing underperforms, R-Pool shows particular promise. This demonstrates the effectiveness of our method with different graph datasets, including those lacking node attributes. Overall, R-Pool demonstrates a good ability to balance clean accuracy and robustness against adversarial attacks, a characteristic not consistently observed in other techniques such as pre-processing.

## 7 Conclusion and Limitations

In this work, we have demonstrated that the choice of pooling operation in graph classification tasks can significantly influence a model's adversarial robustness. Our comprehensive study of three main flat pooling methods revealed that their effectiveness varies across different settings, highlighting the importance of context-specific selection. Building on these theoretical insights, we concluded that filtering nodes after the message-passing scheme but before the pooling operation could enhance robustness. This led to the development of R-Pool, a novel approach that employs Gaussian Mixture Models (GMMs) and an out-of-distribution score to rank nodes and filter out those deemed vulnerable. The proposed method can be adapted to different architectures and doesn't require re-training the model and can directly be employed in the inference time. Through extensive experimental validation across various graph classification datasets, we have demonstrated the efficacy of R-Pool in comparison to existing baselines.

The limitations of this work can be categorized into two main areas. Firstly, our study focused exclusively on flat pooling methods, and therefore extending the analysis to hierarchical pooling represents an important step to demonstrate the universality of our approach. Secondly, the current proposed filtering method, R-Pool, is inherently "post-hoc" in nature. This approach offers a new perspective on the inference time defense, which is important with the current surge of Foundation models. A natural progression of this work would be the development of end-to-end trainable methods that integrate robustness considerations directly into the model's training phase, potentially leading to more adversarial robust graph classifiers.

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

# A  APPENDIX

# B  PROOF OF THEOREM 4.2

**Theorem 4.2** Let $f : (\mathcal{G}, \mathcal{X}) \to \mathcal{Y}$ denote a graph-based function composed of $L$ GCN layers, where the weight matrix of the $i$-th layer is denoted by $W^{(i)}$. Further, let $d^{0,1}$ be a graph distance. For adversarial attacks only targeting node features of the input graph, with a budget $\epsilon$, we have:

- If $f$ is Max-pooling based classifier, then $f$ is $d^{0,1}$-$(\epsilon, \gamma)$ robust with:

$$\gamma = \prod_{l=1}^{L} \|W^{(l)}\| \max_{u \in \mathcal{V}} \hat{w}_u \epsilon$$

- If $f$ is Sum-pooling based classifier, then $f$ is $d^{0,1}$-$(\epsilon, \gamma)$ robust with:

$$\gamma = \prod_{l=1}^{L} \|W^{(l)}\| \sum_{u \in \mathcal{V}} \hat{w}_u \epsilon$$

- If $f$ is Average-pooling based classifier, then $f$ is $d^{0,1}$-$(\epsilon, \gamma)$ robust with:

$$\gamma = \frac{\epsilon}{|V|} \prod_{l=1}^{L} \|W^{(l)}\| \sum_{u \in \mathcal{V}} \hat{w}_u$$

with $\hat{w}_u$ denoting the sum of normalized walks of length $(L-1)$ starting from node $u$.

*Proof.* In this proof, we consider that $f$ is a graph-function that is based on $L$ layers of GCN. We recall taht the GCN message-passing propagation is formulated for a node $u$ as

$$h_u^{(\ell)} = \sigma^{(\ell)} \Big( \sum_{v \in \mathcal{N}(u) \bigcup \{u\}} \frac{W^{(\ell)} h_v^{(\ell-1)}}{\sqrt{(1+d_u)(1+d_v)}} \Big) \tag{2}$$

where $W^{(\ell)} \in \mathbb{R}^{d_{\ell-1} \times d_\ell}$ is the learnable weight matrix with $d_\ell$ being the embedding dimension of layer $\ell$ and $\sigma^{(\ell)}$ is the activation function of $\ell$-th layer. We recall that $h^{(0)} = X \in \mathbb{R}^{n \times d}$ is set to the initial node features.

Similar to the work (ABBAHADDOU et al., 2024), we denote $X$ as the original node features and denote by $X'$ the perturbed adversarial features. We consider a node $u \in V$, we denote by $h_u$ its representation in the clean graph and $h'_u$ its representation in the attacked graph. We consider that the activation functions $(\sigma^{(\ell)})_{1 \leq \ell \leq L}$ are *nonexpensive* (1-Lipschitz continuous). From the work, we have the following result:

$$\|h_u^{(L)} - h'_{u'}^{(L)}\| \leq \prod_{l=1}^{L} \|W^{(l)}\|_2 \| \sum_{v \in \mathcal{N}(u) \bigcup \{u\}} \sum_{j \in \mathcal{N}(v) \bigcup \{v\}} \cdots$$

$$\sum_{z \in \mathcal{N}(y) \bigcup \{y\}} \frac{X_u - X'_u}{\sqrt{(1+d_u)}(1+d_w)(1+d_j) \dots (1+d_y)\sqrt{(1+d_z)}} \|$$

$$\leq \prod_{l=1}^{L} \|W^{(l)}\| \hat{w}_u \epsilon$$

with $\hat{w}_u$ being the sum of normalized walks of length $(L-1)$ starting from node $u$.

The previous results gives us an idea about the behavior of each node's representation when attacked. In the case of graph classification, an additional pooling operation is added, Specifically:

$$\mathbf{h}_{\text{graph}}^{(L)} = \text{Pool}\Big( \{h_u^{(L)}\}_{u \in V} \Big)$$

Hence this proof's goal is to analyze the following quantity:

$$\|\mathbf{h}_{\text{graph}}^{(L)} - \mathbf{h'}_{\text{graph}}^{(L)}\|$$

Let's consider the **Sum-pooling** operation, that can be written as:

$$\mathbf{h}_{\text{graph}}^{(L)} = \sum_{u \in V} h_u^{(L)}$$

We have the following:

$$
\begin{aligned}
\|\mathbf{h}_{\text{graph}}^{(L)} - \mathbf{h}'^{(L)}_{\text{graph}}\| &= \left\| \sum_{u \in V} \left( h_u^{(L)} - h'^{(L)}_u \right) \right\| \\
&\leq \sum_{u \in V} \|h_u^{(L)} - h'^{(L)}_u\| \quad \text{(Triangle Inequality)} \\
&\leq \sum_{u \in V} \left( \prod_{l=1}^{L} \|W^{(l)}\| \right) \|\hat{w}_u\| \epsilon \\
&= \left( \prod_{l=1}^{L} \|W^{(l)}\| \right) \epsilon \sum_{u \in V} \hat{w}_u
\end{aligned}
$$

For the case of the **Average-pooling** operation, that can be written as:

$$\mathbf{h}_{\text{graph}}^{(L)} = \frac{1}{|V|} \sum_{u \in V} h_u^{(L)}$$

We have the following analysis:

$$
\begin{aligned}
\|\mathbf{h}_{\text{graph}}^{(L)} - \mathbf{h}'^{(L)}_{\text{graph}}\| &= \left\| \frac{1}{|V|} \sum_{u \in V} \left( h_u^{(L)} - h'^{(L)}_u \right) \right\| \\
&\leq \frac{1}{|V|} \sum_{u \in V} \|h_u^{(L)} - h'^{(L)}_u\| \quad \text{(Triangle Inequality)} \\
&\leq \frac{1}{|V|} \left( \prod_{l=1}^{L} \|W^{(l)}\| \right) \epsilon \sum_{u \in V} \hat{w}_u
\end{aligned}
$$

In the case of **Max-pooling**:

$$
\begin{aligned}
\left| [\mathbf{h}_{\text{graph}}^{(L)}]_k - [\mathbf{h}'^{(L)}_{\text{graph}}]_k \right| &\leq \max_{u \in V} \left| [h_u^{(L)}]_k - [h'^{(L)}_u]_k \right| \\
&\leq \max_{u \in V} \|h_u^{(L)} - h'^{(L)}_u\| \\
&\leq \left( \prod_{l=1}^{L} \|W^{(l)}\| \right) \max_{u \in V} \hat{w}_u \epsilon
\end{aligned}
$$

By taking into account the expectancy (as shown in Definition 1), we get the desired results.

$$\square$$

## C  PROOF OF THEOREM 4.3

**Theorem 4.3** Let $f : (\mathcal{G}, \mathcal{X}) \to \mathcal{Y}$ be composed of $L$ GIN-layers (with its internal parameter $\zeta = 0$) and let $W^{(i)}$ denote the weight matrix of the $i$-th MLP layer. We consider the input node feature space to be bounded i. e., $\|X\|_2 < B$ for some $B \in \mathbb{R}$. For node feature-based attacks, with a budget $\epsilon$, the function $f$ is $(d^{0,1}, \epsilon)$–robust with

- If $f$ is Max-pooling based classifier, then $f$ is $d^{0,1}$-$(\epsilon, \gamma)$ robust with:

$$\gamma = \prod_{l=1}^{L} \|W^{(l)}\| [B \times L \times \max_{u \in V} \deg(u) + \epsilon]$$

- If $f$ is Sum-pooling based classifier, then $f$ is $d^{0,1}$-$(\epsilon, \gamma)$ robust with:

$$\gamma = \prod_{l=1}^{L} \left\|W^{(l)}\right\| [2B \times L \times |E| + |V|\epsilon]$$

- If $f$ is Average-pooling based classifier, then $f$ is $d^{0,1}$-$(\epsilon, \gamma)$ robust with:

$$\gamma = \prod_{l=1}^{L} \left\|W^{(l)}\right\| \left[\frac{2B \times L \times |E|}{|V|} + \epsilon\right]$$

with $|E|$ being the number of edges and $|V|$ the number of nodes.

*Proof.* In this proof, we consider that $f$ is based on $L$ GIN-layers (with a parameter $\zeta = 0$, usually denoted as $\epsilon$). The GIN message-passing propagation process can be written for a node $u$ as:

$$h_u^{(\ell+1)} = T^{(\ell+1)}((1+\zeta)h_u^{(\ell)} + \sum_{v \in \mathcal{N}(u)} h_v^{(\ell)})$$

with $T$ denoting a Neural Networks (a MLP) for example and $\zeta$ denotes the parameter of the GIN. We recall that $h^{(0)} = X \in \mathbb{R}^{n \times d}$ is set to the initial node features.

Similar to the previous proof, we base our proof on previous work (ABBAHADDOU et al., 2024), we denote $X$ as the original node features and denote by $X'$ the perturbed adversarial features. We consider a node $u \in V$, we denote by $h_u$ its representation in the clean graph and $h'_u$ its representation in the attacked graph. We consider that the activation functions $(\sigma^{(\ell)})_{1 \le \ell \le L}$ are *nonexpensive* (1-Lipschitz continuous).

We use the same assumptions as the one considered in (ABBAHADDOU et al., 2024). Specifically, we consider that the input feature space $\mathcal{H}_0$ is bounded, thus each hidden space $\mathcal{H}_i$ of the iterative process of message passing is bounded and let $B = \max_{\ell \le L} B_\ell$ be its global maximum bound. We additionally consider that GIN-parameter $\zeta \approx 0$ (which is very frequent in the literature). We have therefore the following result:

$$\|h_u^{(\ell+1)} - h'^{(\ell+1)}_{u'}\| \le \prod_{l=1}^{L} \|W^{(l)}\|[B \times L \times deg(u) + \epsilon]$$

From this perspective, let's consider the case of graph classification, where we start by the **sum-pooling** operation:

$$\left\|h_G^{(L)} - h'^{(L)}_G\right\| = \left\|\sum_{u \in V} \left(h_u^{(L)} - h'^{(L)}_u\right)\right\|$$

$$\le \sum_{u \in V} \left\|h_u^{(L)} - h'^{(L)}_u\right\| \quad \text{(by the triangle inequality)}$$

$$\le \prod_{l=1}^{L} \left\|W^{(l)}\right\| \sum_{u \in V} [B \times L \times \deg(u) + \epsilon]$$

$$= \prod_{l=1}^{L} \left\|W^{(l)}\right\| \left[B \times L \times \sum_{u \in V} \deg(u) + |V|\epsilon\right].$$

In the case of undirected graph, since $\sum_{u \in V} \deg(u) = 2|E|$, we have:

$$\left\|h_G^{(L)} - h'^{(L)}_G\right\| \le \prod_{l=1}^{L} \left\|W^{(l)}\right\| [2B \times L \times |E| + |V|\epsilon]$$

Similar for the case of **Average-pooling**, we have:

$$\left\|h_G^{(L)} - h'_G^{(L)}\right\| = \frac{1}{|V|}\left\|\sum_{u \in V}\left(h_u^{(L)} - h'_u^{(L)}\right)\right\|$$

$$\leq \frac{1}{|V|}\sum_{u \in V}\left\|h_u^{(L)} - h'_u^{(L)}\right\|$$

$$\leq \prod_{l=1}^{L}\left\|W^{(l)}\right\|\left[\frac{2B \times L \times |E|}{|V|} + \epsilon\right].$$

In the case of **Max-pooling** operation, we have the following:

$$\left\|h_G^{(L)} - h'_G^{(L)}\right\| \leq \prod_{l=1}^{L}\left\|W^{(l)}\right\|\left[B \times L \times \max_{u \in V}\deg(u) + \epsilon\right].$$

By taking into account the expectancy (as shown in Definition 1), we get the desired results.

# D   TIME COMPLEXITY ANALYSIS

As explained in Section 5, the main computational complexity of our method is concentrated in the EM algorithm used for estimating the GMM's parameters. The EM algorithm is an iterative process and hence the complexity mainly depends on the number of iterations that have been chosen. For our experimentation, we have seen that 100 iterations was a good number to reach a satisfactory accuracy. In table 2, we provide a time complexity comparison of our R-Pool to other considered baselines on the used graph classification datasets.

Table 2: Mean training time analysis (in s) of a our R-Pool in comparison to the other considered benchmarks on the graph classification datasets.

| DATASET | GCN | RANDOMZIED SMOOTHING | PRE-PROCESSING | R-POOL |
|---|---|---|---|---|
| PROTEINS | 0.001 | 0.014 | 0.037 | 0.013 |
| NCI1 | 0.008 | 0.019 | 0.01 | 0.015 |
| IMDB-BINARY | 0.0007 | 0.013 | - | 0.011 |

# E   EXPERIMENTAL DETAILS

## E.1   DATASETS

For our experimentation, we mainly used the classical graph datasets derived from bioinformatics and chemoinformatics (PROTEINS, NCI1) and social networks (IMDB-BINARY) (Morris et al., 2020). We used the public folds and the experimental setting that was provided by the work Errica et al. (2020). Details about the dataset are provided in Table 3.

## E.2   IMPLEMENTATION DETAILS

Our implementation is available in the supplementary materials (and will be publicly available afterwards). It is built using the open-source library *PyTorch Geometric* (PyG) under the MIT license (Fey & Lenssen, 2019) and DGL in the case of the genetic and random attacks. We leveraged the publicly available implementation of both the attacks and the benchmarks. The experiments have been run on both a NVIDIA A100 GPU.

For all the attacks, we set the number of attack epochs to 100 and in a "re-wiring" mode, meaning that the attack can either add/delete an edge.

Table 3: Statistics of the graph classification datasets used in our experiments.

| DATASET | #GRAPHS | #NODES | #EDGES | #CLASSES |
|---|---|---|---|---|
| IMDB-BINARY | 1000 | 19.77 | 96.53 | 2 |
| NCI1 | 4110 | 29.87 | 32.30 | 2 |
| PROTEINS | 1113 | 39.06 | 72.82 | 2 |

For all our experiments, we employed a 2-layer convolutional architecture (consisting of two iterations of message passing and updating) stacked with a Multi-Layer Perception (MLP) as a readout using the Adam Optimizer Kingma & Ba (2015). We train the model for 100 epochs (as this was sufficient to reach the state-of-the art accuracy for these models) and we use a learning rate of $1e-02$.

$\square$

