# OpenReview forum: "Adversarially Robust Graph Classification: A Pooling-Based Defense Framework"
_ICLR.cc/2025/Conference — ICLR 2025 Conference Withdrawn Submission_

### Official Review · Reviewer_79Rs · 2024-11-03

**Soundness:** 2
**Presentation:** 3
**Contribution:** 1
**Rating:** 3
**Confidence:** 3

**Summary:**

This paper focuses on graph classification and inference-time defenses. The authors have demonstrated that the choice of pooling operation in graph-level tasks can significantly influence a model’s adversarial robustness. Based on this analysis, The authors propose a pre-pooling operation, called R-Pool (Robust-Pooling), which is based a filtering mechanism using Gaussian Mixture Models (GMMs) to detect and exclude nodes heavily impacted by attacks, thereby enhancing robustness at inference time.

**Strengths:**

1.  The structure of the paper is clear and easy to follow.
2.  This paper explores the graph-classification robustness problem, which has been neglected in graph adversarial robustness.
3.  This method can be used with any pooling operation and any underlying model, and does not require re-training the model nor adapting its architecture.

**Weaknesses:**

1. The novelty seems limited. The core idea of R-Pool is to mitigate the impact of these perturbations by filtering out such nodes in the graph that accumulate adversarial perturbation information. However, the GEM score is similar to the OOD detection method[1]. The technical contribution is a little weak.
2. The experiments in this paper are not sufficient. There are only three datasets and the datasets used in experiments only have two classes. To further verify the effectiveness of the method on a wide range of datasets, I suggest that the author conduct experiments on datasets  with more classes.
3.  It is recommended that the authors give some results of graph pooling methods in adversarial robustness, such as max-pooling, sum-pooling and average-pooling, to demonstrate the effectiveness of  R-Pool.
4.  From Table 1 in the experimental section, it can be seen that in many cases, the proposed R-Pool has worse performance than some baseline methods, which makes it difficult to illustrate the effectiveness of the method.

[1] Provable guarantees for understanding out-of-distribution detection. AAAI, 2022.

**Questions:**

See the weaknesses.

---

### Official Review · Reviewer_TD6D · 2024-11-03

**Soundness:** 2
**Presentation:** 2
**Contribution:** 1
**Rating:** 3
**Confidence:** 4

**Summary:**

This paper focuses on the robustness of graph neural networks on graph classification tasks. Specifically, this paper proposes to revise the pooling (readout) operation to improve the robustness against the adversarial attacks on graph neural networks. This robust pooling is base on a filtering mechanism with Gaussian Mixture Models to detect nodes affected by malicious edges and attributes. Experiments are conducted on three small datasets. No baselines are compared in the experiments.

**Strengths:**

1. The literature review is conducted comprehensively, which makes the readers easy to follow the background.
2. The preliminary knowledge and definition of adversarial robustness are extensively introduced, which is very clear.

**Weaknesses:**

1. The experiments are conducted poorly. There are extensive methods in defending adversarial attacks on graph neural networks. Many of  them such as the Soft Median [1], MedianGNN [2] and GCN-jaccard [3]  can be easily applied in the graph classification. However, no existing work is compared in the work. Hence, the superiority of the proposed method cannot be justified.
2. Experiments are only conducted on datasets with small sizes. In addition, the proposed method is only applied to the GCN. It is required to investigate the ability in improving the robustness on large-scale datasets and more diverse model architectures.
3.  Though the proposed method is very simple, the methodology is written poorly, making the reviewer difficult to follow some details of the method. For example, how the score in line 343 is used? Is it simply used to filter out the nodes? Or is the score also used for the pooling?
4. The technical contribution of the proposed method is limited. The major idea is to identify and eliminate the OOD nodes. Similar method that eliminating the malicious nodes have been well investigated in Soft Median [1], MedianGNN [2] and GCN-Jaccard [3].

[1] Geisler, Simon, et al. "Robustness of graph neural networks at scale." Advances in Neural Information Processing Systems 34 (2021): 7637-7649.

[2] Chen, Liang, et al. "Understanding structural vulnerability in graph convolutional networks." arXiv preprint arXiv:2108.06280 (2021).
[3] Wu, Huijun, et al. "Adversarial examples on graph data: Deep insights into attack and defense." arXiv preprint arXiv:1903.01610 (2019).

**Questions:**

Please refer to the weakness.

---

### Official Review · Reviewer_Sqt9 · 2024-11-04

**Soundness:** 2
**Presentation:** 3
**Contribution:** 2
**Rating:** 5
**Confidence:** 5

**Summary:**

This work investigates adversarial robustness in graph classification tasks. The authors start with defining adversarial robustness and upper bounds with different pooling operators. Through theoretical analysis, they argue that the key to improving robustness lies in reducing the impact of attacked nodes, which are often those with high degrees. Therefore, this objective can be achieved by discarding highly connected nodes. Specifically, the authors propose a pooling strategy based on Gaussian mixture-based energy measurement (GEM) to filter out these potentially vulnerable nodes.

**Strengths:**

1. The writing is clear and the logic flow is easy to follow.

2. The theoretical part of this work is sound and easy to understand.

**Weaknesses:**

### **There is very little research on the robustness of graph classification tasks. As the authors mentioned, previous work has primarily focused on node classification tasks, therefore:**

1. What are the practical and theoretical implications of studying robustness in graph classification tasks? For example, node classification has highly relevant applications such as financial fraud detection. What practical scenarios support the study of graph classification? If not, what contributions does researching the adversarial robustness of graph classification provide for our understanding of GNNs?

2. The premise of studying adversarial robustness is to have a reasonable adversarial attack in place. Although PGD can be directly adapted to graph classification, there are still some details worth discussing. For instance, PGD sets the budget such that the total perturbation cannot exceed a certain proportion of the graph structure, whereas this work defines a new distance metric to ensure the attack is unnoticeable. The authors should provide more details on how this unnoticeability is integrated into the PGD.

### **The idea of disregarding high-degree nodes need to be justified**

On one hand, degree-aware GNNs (for both node and graph classification) have existed for quite some time. On the other hand, previous research generally suggests that gradient-based attaks like PGD are more likely to attack low-degree nodes, so the choice in this work to discard high-degree nodes needs further justification. If possible, could the authors provide some experiments to demonstrate that high-degree nodes are more significantly affected by attacks?

[1] Wu, Jun, Jingrui He, and Jiejun Xu. "Net: Degree-specific graph neural networks for node and graph classification." Proceedings of the 25th ACM SIGKDD international conference on knowledge discovery & data mining. 2019.

[2] Gosch, Lukas, et al. "Revisiting robustness in graph machine learning." arXiv preprint arXiv:2305.00851 (2023).

[3] Li, Kuan, et al. "Revisiting graph adversarial attack and defense from a data distribution perspective." The Eleventh International Conference on Learning Representations. 2023.

**Questions:**

See weaknesses

---

### Official Review · Reviewer_vEsv · 2024-11-05

**Soundness:** 3
**Presentation:** 3
**Contribution:** 2
**Rating:** 5
**Confidence:** 3

**Summary:**

This work introduces a novel inference-time defense for Graph Neural Networks (GNNs) in graph classification tasks, proposing R-Pool, a pre-pooling operation that enhances robustness by detecting and excluding nodes impacted by adversarial attacks using Gaussian Mixture Models. Unlike existing methods, this approach requires no model retraining or architectural changes and demonstrates significant improvements in robustness across various adversarial scenarios.

**Strengths:**

1. The paper is well organized. I like the way the authors present the challenges and the addressing steps in the introduction, which make the paper easy to read and understand.
2. The authors conduct ablation studies to evaluate the performance of the proposed modules, and several experiments are conducted to test the sensitivity of hyperparameters.
3. The proposed method has theory support.

**Weaknesses:**

1. The attack methods are relatively simple and old. Authors may consider more advanced attack methods such as node injection attacks [1].

2. The baselines are limited. I'm curious about the performance of the proposed method compared with the previous defense method [2,3 ,4], or the performance of the proposed method as a plug-and-play module on other defense methods.

3. Authors need to discuss these recent attack and defense methods [1,2,3,4] in related works.

[1] Graph robustness benchmark: Benchmarking the adversarial robustness of graph machine learning, NeurIPS 2021
[2] Robustness of Graph Neural Networks at Scale, NeurIPS 2021
[3] GARNET: Reduced-rank topology learning for robust and scalable graph neural Networks, Learning on Graphs Conference 2022
[4] Mitigating Emergent Robustness Degradation while Scaling Graph Learning

**Questions:**

see weaknesses

---

### Note · Authors · 2024-12-04

**Comment:**

We would very much like to thank the reviewers for their insightful reviews and the AC for taking the time to handle our paper. Unfortunately, we were unable to produce a comprehensive, convincing rebuttal during the discussion period and would therefore like to withdraw the paper to not take anymore of your time. We will certainly take the valuable feedback provided into account in future versions of this work. Thank you very much again.

**Withdrawal Confirmation:**

I have read and agree with the venue's withdrawal policy on behalf of myself and my co-authors.